# Mutation of an Essential 60S Ribosome Assembly Factor MIDASIN 1 Induces Early Flowering in *Arabidopsis*

**DOI:** 10.3390/ijms23126509

**Published:** 2022-06-10

**Authors:** Ke Li, Pengfei Wang, Tingting Ding, Lei Hou, Guanghui Li, Chuanzhi Zhao, Shuzhen Zhao, Xingjun Wang, Pengcheng Li

**Affiliations:** 1Shandong Provincial Key Laboratory of Crop Genetic Improvement, Ecology and Physiology, Institute of Crop Germplasm Resources, Shandong Academy of Agricultural Sciences, Jinan 250100, China; like_sdu@foxmail.com (K.L.); houlei9042@163.com (L.H.); ghli66@126.com (G.L.); chuanzhiz@126.com (C.Z.); zhaoshuzhen51@126.com (S.Z.); xingjunw@hotmail.com (X.W.); 2Shandong Academy of Grape, Jinan 250100, China; fengqiaoyouzi@126.com (P.W.); rainbowisting@163.com (T.D.)

**Keywords:** *Arabidopsis*, *MDN1*, ribosome biogenesis defect, early flowering, ABI5

## Abstract

Ribosome biogenesis is tightly associated with plant growth and reproduction. Mutations in genes encoding ribosomal proteins (RPs) or ribosome biogenesis factors (RBFs) generally result in retarded growth and delayed flowering. However, the early-flowering phenotype resulting from the ribosome biogenesis defect is rarely reported. We previously identified that the AAA-ATPase MIDASIN 1 (MDN1) functions as a 60S RBF in *Arabidopsis*. Here, we found that its weak mutant *mdn1-1* is early-flowering. Transcriptomic analysis showed that the expression of *FLOWERING LOCUS C* (*FLC*) is down-regulated, while that of some autonomous pathway genes and *ABSCISIC ACID-INSENSITIVE 5* (*ABI5*) is up-regulated in *mdn1-1*. Phenotypic analysis revealed that the flowering time of *mdn1-1* is severely delayed by increasing *FLC* expression, suggesting that the early flowering in *mdn1-1* is likely associated with the downregulation of *FLC*. We also found that the photoperiod pathway downstream of *CONSTANTS* (*CO*) and *FLOWERING LOCUS T* (*FT*) might contribute to the early flowering in *mdn1-1*. Intriguingly, the *abi5-4* allele completely blocks the early flowering in *mdn1-1*. Collectively, our results indicate that the ribosome biogenesis defect elicited by the mutation of MDN1 leads to early flowering by affecting multiple flowering regulation pathways.

## 1. Introduction

The eukaryotic 80S ribosome consists of two subunits, the 60S large subunit, and the 40S small subunit. The 60S subunit comprises 25S, 5.8S, and 5S rRNA and approximately 47 RPs, whereas the 40S subunit comprises 18S rRNA and approximately 33 RPs [1,2]. Ribosome biogenesis is a fundamental process. Based on the research on yeast and animal there are hundreds of RBFs involved in the pre-rRNA transcription, modification, processing, folding, and the incorporation of RPs [2]. Ribosome biogenesis begins with the transcription of the 45S rRNA precursor (pre-rRNA) by RNA polymerase I, and the 45S pre-rRNA functions as a platform for the 90S particle assembly [3,4]. MIDASIN1 (MDN1), an AAA-ATPase, is conserved in different species. MDN1 successively interacts with the nucleolus protein Ytm1/PES2 and the nucleoplasm protein NOTCHLESS (NLE) through its C-terminal metal ion-dependent adhesion site (MIDAS) domain, to trigger these proteins’ release from the pre-60S particle [5,6,7,8]. The ribosome is the main component of the protein synthesis machinery and is essential for normal cell growth, and defects in RBFs and RPs always exert pleiotropic effects on cell proliferation and growth [4,9]. In *Arabidopsis*, the ribosome biogenesis defect generally causes abnormal embryogenesis, seed germination, leaf morphology, retarded root elongation, and delayed flowering [4,6,7,9,10]. These findings imply ribosome-mediated regulation roles throughout the entire plant life cycle. In the face of endogenous and environmental stresses, the most urgent task for plants is to survive and to reproduce the next generation, which is tightly associated with the flowering time. Therefore, elucidating the mechanism of flowering time regulation under the ribosome biogenesis defect is helpful for us to understand ribosome-mediated regulation roles.

Flowering is one of the most studied developmental processes in the plant over the last 30 years. A large number of studies have provided a good understanding of how endogenous and environmental cues precisely control the switch from vegetative growth to reproductive growth [11,12,13]. In winter-annual accessions of *Arabidopsis*, *FRIGIDA* (*FRI*) activates the expression of the flowering suppressor *FLOWERING LOCUS C* (*FLC*) to a high level by enrichment of active epigenetic modification at the *FLC* locus and leads to late flowering [14,15,16]. The vernalization pathway triggers flowering by directly repressing *FLC* expression by enrichments of repressive epigenetic modification at the *FLC* locus [17,18]. In contrast, rapid-cycling accessions of *Arabidopsis* contain a loss-of-function mutation in *FRI* and therefore have a low *FLC* level and are early flowering without vernalization [16]. Loss-of-function mutants of autonomous pathway genes in rapid cycling background display a similar late-flowering phenotype as the functional *FRI* containing winter annuals [13,14,19]. Those autonomous pathway genes repress the *FLC* expression also by enrichments of repressive epigenetic modification at the *FLC* locus [13]. The autonomous pathway proteins include FLOWERING LOCUS D (FLD, H3K4 demethylase), FLOWERING LOCUS VE (FVE, recruits histone deacetylase complex), LUMINIDEPENDENS (LD, a homeodomain-containing protein), FLOWERING CONTROL LOCUS A (FCA), and FPA (with the RNA-binding motifs that function partly through FLD to mediate *FLC* locus), FLOWERING LOCUS KH DOMAIN (FLK) and FY (RNA-binding and RNA-processing proteins) [20,21,22,23,24,25,26]. Polycomb group (PcG) proteins function as transcriptional repressors of developmental gene expression and Polycomb repressive complex2 (PRC2) catalyzes repressive histone 3 Lys-27 trimethylation (H3K27me3) [27,28]. In *Arabidopsis*, CURLY LEAF (CLF) is one of the homologs of the *Drosophila* H3K27 methyltransferase Enhancer of Zeste [E(z)] and plays an important role in the developmental switch from the vegetative phase to reproduction [29]. CONSTANTS (CO) acts as a flowering activator that promotes the expression of *FLOWERING LOCUS T* (*FT*) and mediates the photoperiod pathway in long-day conditions, whereas in short-day conditions, CO protein is not being stably produced and might not function in the control of flowering [30,31,32,33,34]. CLF catalyzes H3K27 trimethylation on both the *FLC* and *FT* locus to repress their expression [35]. Consequently, CO and FLC, as two central flowering regulators, CO and FLC, antagonistically regulate flowering [31,36]. Their downstream flowering pathway integrators, such as FT, SUPPRESSOR OF OVEREXPRESSION OF CO 1 (SOC1), and LEAFY (LFY), can integrate signals from multiple flowering pathways and determine the flowering time depending on their expression levels [37,38,39].

Abscisic acid (ABA) has been reported to play both positive and negative roles in flowering time regulation [40]. ABA biogenesis-related mutants in the rapid-cycling background (Col-0, *aba1*, *aba1-6*, *aba2-1*, and *aba2-4*) of *Arabidopsis* are late flowering under long-day conditions, whereas ABA insensitive mutants (*abi1-1*, *abi4-1*, and *abi5-4*) are early flowering [41,42,43,44]. Endogenous ABA promotes *FT* expression, meanwhile, ABSCISIC ACID-INSENSITIVE 4 (ABI4) and ABI5, two positive ABA signaling regulators, can directly bind to the *FLC* promoter and activate its transcription [41,42].

Generally, defects in ribosome biogenesis lead to retarded cell division and elongation, and an associated late-flowering phenotype [10,45,46,47,48,49,50]. Here, we described that deficiency in the *MDN1* function elicits an unexpected early-flowering phenotype by affecting multiple flowering regulation pathways.

## 2. Materials and Methods

### 2.1. Plant Materials and Growth Conditions

*Arabidopsis* accession Col-0 was used in this study. The *mdn1-1* mutant was screened in our lab as described previously [6,7]. T-DNA insertion mutant *mdn1-2* (Salk_057010) was obtained from the *Arabidopsis* Biological Resource Center. To generate *mdn1-1*/+ *mdn1-2*/+, *mdn1-1* was crossed with *mdn1-2*/+ and genotyped with specific primers (Appendix A). The *flc-3* (104 bp deletion in the first exon), *flk-4* (Salk_112850), *fve-4* (TGG to TGA mutation in the 6th exon), *co* (SAIL_24_H04), *ft-10* (GABI_290E08), and *clf-28* (SALK_139371) mutants were kindly provided by prof. Scott Michaels (Indiana University Bloomington). The double mutant of *mdn1-1 abi5* had been described in our previous study [51]. Seedlings and plants were grown under 16 h light/8 h dark (LD, long day) or 8 h light/16 h dark (SD, short day) conditions at 21 °C.

### 2.2. Quantification of Flowering Time

Seedlings and plants were grown to flowering in the growth chamber under LD or SD condition. Flowering time was determined by numbering total rosette leaves at bolting and the days from germination to flowering from three biological replicates.

### 2.3. RNA Sequencing and RNA-seq Data Analysis

Total RNA was extracted from 7 DAG seedling aerial parts grown at 21 °C under LD conditions using the RNAiso (Takara) and treated with DNase I (Takara) to remove genomic DNA according to the manufacturer’s protocol as described previously [52]. RNA quality of all samples was detected by Agilent 2100 Bio analyzer (Agilent RNA 6000 Nano Kit) and NanoDrop. The mRNA was enriched by the Oligo dT Selection method and cleaved into short fragments. Fragmented the RNA and reversed transcription to double-strand cDNA (dscDNA) by N6 random primer. The synthesized cDNA was subjected to end-repair and then was 3′ adenylated. Adaptors were ligated to the ends of these 3′ adenylated cDNA fragments. The ligation products were purified, and many rounds of PCR amplification were performed to enrich the purified cDNA template using PCR primer. Denature the PCR product by heat and the single-strand DNA is cyclized by splint oligo and DNA ligase to construct the cDNA library. Three biological replicates of each sample were used for RNA-Seq on the BGISEQ-500 platform by Beijing Genomics Institute (BGI). The RNA-seq raw sequencing data have been submitted in the SRA database under BioProjects: PRJNA564539 (https://www.ncbi.nlm.nih.gov/sra/PRJNA564539, 7 October 2020).

The low-quality reads with adaptors and reads with unknown bases (N bases more than 5%) were filtered to get the clean reads using SOAPnuke software. All clean reads were aligned to the *Arabidopsis* TAIR10 reference genome using Bowtie2 (http://bowtie-bio.sourceforge.net/Bowtie2/index.shtml) and calculated gene expression level with RSEM (http://deweylab.biostat.wisc.edu/RSEM). Gene expression level was normalized by using the FPKM (Fragments Per Kilobase of transcript per Million mapped reads) method. Differentially expressed genes (DEGs) were detected using DEGseq software (Fold Change ≥ 2 and Adjusted *p*-value ≤ 0.001) and used Volcano Plot to show the summary of DEGs. KEGG (Kyoto Encyclopedia of Genes and Genomes) pathway was classified according to official classification, and the significantly enriched KEGG pathways were identified using a hypergeometric test under the standard of *p*-value ≤ 0.01.

### 2.4. Analysis of Transcript Abundance

Total RNA was extracted from 5 DAG seedling aerial parts grown at 21 °C under LD conditions using the RNAiso (Takara). Using DNase I treatment to remove genomic DNA and 5 µg of total RNA was used for first-strand cDNA synthesis (Takara) according to manufacturer’s protocol. *Actin-2* was used as a control gene for qRT-PCR and the relative expression of genes was calculated by ABI7500 Real-Time PCR System using the 2^−^^△△^^Ct^ method.

## 3. Results

### 3.1. Mutation in MDN1 Leads to Early Flowering

Previous studies showed a common late-flowering phenotype in *Arabidopsis* ribosome-deficient mutants (Appendix A). To know if the mutation of *MDN1* influences flowering time, we compared this phenotype between *mdn1-1* and wild-type (WT) plants. Intriguingly, the *mdn1-1* plants flowered about 2 and 8.5 days earlier than WT under long-day conditions (LD, 16 h of light and 8 h of darkness at 21 °C) and short-day conditions (SD, 8 h of light and 16 h of darkness in 21 °C), respectively (Figure 1A,B,D,E). The early-flowering phenotype of *mdn1-1* was also testified by numbering the rosette leaves at the first flower bud bolting. *mdn1-1* plants produced fewer rosette leaves under both LD (9 vs. 15) and SD (34 vs. 49) conditions than WT (Figure 1C,F). To confirm that the early-flowering phenotype was caused by the *mdn1* mutation, we crossed *mdn1-1* with a heterozygous T-DNA insertion mutant *mdn1-2*/+ (Salk_057010). We found that *mdn1-1*/*mdn1-2* showed a similar early-flowering phenotype with *mdn1-1,* suggesting that mutation in *MDN1* led to early flowering (Appendix A). These observations indicate a difference between *mdn1-1* and other known ribosome-associated mutants in timing the reproductive growth, which led us to explore the mechanism underlying the early-flowering phenotype of *mdn1-1*.

### 3.2. Multiple Flowering Pathways Are Influenced in mdn1-1

To gain insight into the global transcription responses of the *MDN1* mutation, we performed transcriptome profiles analysis of the aerial part from 7 DAG seedlings (under LD conditions). A total of 2049 differentially expressed genes (DEGs, |log_2_(fold-change)| ≥ 1 and *p*-value < 0.001) were identified in *mdn1-1*. Among these genes, 922 were up-regulated, and 1127 were down-regulated (Appendix A).

To gain more information on flowering time regulation, gene expression of flowering time regulators was analyzed (Figure 2). Importantly, autonomous pathway genes (*LD*, *REF6*, *FY*, *FPA*, *FCA*, *FLK,* and *FVE*) were up-regulated, and *FLC* was down-regulated (Figure 2A). The expression of *ABI5* and *ABSCISIC ACID RESPONSIVE ELEMENTS-BINDING FACTOR 3* (*ABF3*) was up-regulated (Figure 2A). The expression of photoperiod pathway-related genes, including *FT*, *CRYPTOCHROMEs* (*CRY1* and *CRY2*), and *PHYTOCHROMEs* (*PHYA* and *PHYC*), was down-regulated (Figure 2A). In addition, the expression of *TIMING OF CAB OF EXPRESSION 1* (*TOC1*), *LHY/CCA1-LIKE 1* (*LCL1*), *ELONGATED HYPOCOTYL 5* (*HY5*), and *LONG HYPOCOTYL IN FAR-RED 1* (*HFR1*) was also down-regulated (Figure 2A). Decreased *FLC* and *FT* expression and increased *ABI5*, *FLK*, *FVE,* and *LD* expression in the *mdn1-1* mutant were confirmed by qRT-PCR (Figure 2B). The above observations suggest that the *MDN1* mutation produces a global influence on the expression of flowering time regulator genes.

### 3.3. Downregulation of FLC Is Associated with Early Flowering in mdn1-1

FLC is a repressor of flowering, and down-regulation of the *FLC* mRNA level results in early flowering under both LD and SD conditions in some *Arabidopsis* mutants [14,53]. To test the involvement of *FLC* in *mdn1-1* early flowering, the *flc-3* allele was introduced into the *mdn1-1* background. Under LD conditions, the flowering time of *flc-3* was similar to *mdn1-1*, which showed about 2 d earlier than WT (Figure 3A,B). In addition, the number of rosette leaves of *flc-3* was about four less than WT and three more than that of *mdn1-1* (Figure 3C). Unexpectedly, the flowering time of the *mdn1-1 flc-3* double mutant was later than both single mutants (Figure 3A–C). Meanwhile, the number of rosette leaves of *mdn1-1 flc-3* was more than that of *flc-3* and *mdn1-1* (Figure 3C). To test the effect of improving *FLC* expression on *mdn1-1* flowering time, two weak autonomous pathway mutants, *flk-4* and *fve-4*, were employed. The expression of *FLC* was up-regulated in both *mdn1-1 flk-4* and *mdn1-1 fve-4* double mutants (Appendix A). Phenotypic analysis showed that both *flk-4* and *fve-4* could significantly delay the flowering time in *mdn1-1* (Figure 3D–F), suggesting that the early-flowering phenotype of *mdn1-1* might be associated with the downregulated *FLC*.

### 3.4. Photoperiod Pathway Contributes to mdn1-1 Early Flowering

Plants sense upcoming environmental changes by detecting photoperiod information and precisely control the timing of flowering at a suitable condition. To investigate the role of photoperiod pathway regulators in *mdn1-1* flowering time, we also introduced *co* and *ft-10* mutations into the *mdn1-1* mutant background. Under LD condition, the *mdn1-1 co* double mutant showed about a 5-d acceleration of flowering than *co* and about 28-d repression than *mdn1-1* (Figure 4A,C). Correspondingly, the number of rosette leaves of *mdn1-1 co* was about 4 less than that of *co* and about 26 more than that of *mdn1-1* (Figure 4A,D). The *mdn1-1 ft-10* double mutant showed about a 23-d acceleration of flowering than *ft-10* and about 9-d repression than *mdn1-1* (Figure 4B,C). Meanwhile, the number of rosette leaves of *mdn1-1 ft-10* was about 17 less than that of *ft-10* and was about 12 more than that of *mdn1-1* (Figure 4A,D). *mdn1-1 co* and *mdn1-1 ft-10* plants flowered earlier than *co* and *ft-10*, respectively, and both later than the *mdn1-1* plants (Figure 4B,C), indicating that the early flowering in *mdn1-1* is partially dependent on the photoperiod pathway.

In the *clf-28* plant, the expression of *FLC* and *FT* is de-repressed, while the overexpression of *FT* bypasses the FLC-dependent pathway, which results in an early-flowering phenotype in the *clf-28* plant [35]. We introduced *clf-28* mutation into the *mdn1-1* mutant background. *mdn1-1 clf-28* double mutant flowered earlier than *mdn1-1*, but later than *clf-28* under LD conditions. This observation further supports the above hypothesis (Figure 5).

### 3.5. ABI5 Contributes to Early Flowering in mdn1-1

Since ABA signaling regulates plant flowering and *ABI5* was overexpressed in *mdn1-1*, we wonder the role of *ABI5* in the alteration of *mdn1-1* flowering time. To address this issue, we employed the double mutant of *mdn1-1 abi5-4* which had been described in our previous study [51]. Under both LD and SD condition, *abi5-4* showed a similar flowering time compared with WT in our experimental conditions. Intriguingly, the early-flowering phenotype of *mdn1-1* was completely blocked by the *abi5-4* allele, and the *mdn1-1 abi5-4* double mutant even bolted later than both WT and *abi5-4*. The flowering time of *mdn1-1 abi5-4* double mutant showed about 8 and 25 d later than that of *mdn1-1* under LD and SD conditions, respectively (Figure 6A–C,E). In addition, the number of rosette leaves of this double mutant was about 6 and 21 more than that of *mdn1-1* under LD and SD conditions, respectively (Figure 6D,F). These results suggest that ABI5 positively regulates flowering in *mdn1-1*.

## 4. Discussion

Understanding the mechanism by which plants coordinate their vegetative and reproductive growth at fitness time is an important paradigm in plant biology. However, our understanding of how plants coordinate the vegetative and reproductive growth in response to ribosome deficiency is limiting. Ribosome biogenesis is a complex process that requires the coordinated function of RNA polymerases, hundreds of RBFs, and RPs to facilitate rRNA maturation and RP incorporation, and any dysfunction would significantly influence the plant development [1,9,54]. Emerging studies highlighted the existence of a signaling pathway in ribosome deficiency-induced plant abnormal development, but the detailed mechanism is lacking due to the multiple phenotype outputs [9,54]. The flowering time may serve as a model for us to explore how plant adaptation responds to ribosome deficiency.

In *Arabidopsis*, the phenotypes of mutants with defects in RBFs and RPs have been widely described [4,10,54]. Mutants related to *Arabidopsis* ribosome biogenesis from previous studies showed a common late-flowering phenotype (Appendix A). Most of these previous studies focused on rRNA modification and processing defects and ribosomal protein mutations. Generally, ribosome biogenesis defects triggered by these mutations can be compensated by activating alternative rRNA processing pathways or replacing them by other ribosomal protein members in plants (Appendix A) [4,10]. For example, two orthologues of *LSG1*, *LSG1-1,* and *LSG1-2*, have been identified in *Arabidopsis*, and only LSG1-2 physically interacts with the ribosome and functions in the final step of pre-40S maturation [55]. The homozygous T-DNA insertion lines of *LSG1-1* and *LSG1-2* are viable with normal flowering time, whereas flowering is drastically delayed in *lsg1-1+/– lsg1-2–/–* (Appendix A) [55].

However, the *MDN1* mutation led to an early-flowering phenotype (Figure 1). This contradiction might be associated with MDN1 functions in essential ribosome biogenesis and transport checkpoints (pre-ribosome transport from the nucleolus to the nucleoplasm and from nucleoplasm to cytoplasm) [3,6]. At the same time, MDN1 itself might have a specific role in regulating the reproductive growth transition. To clarify the mechanism of the early flowering in *mdn1-1*, global gene expression profiles of the aerial part of 7 DAG seedlings were employed. The expression of *FLC* was down-regulated in *mdn1-1* (Figure 2), which contributed to the early-flowering phenotype to some extent (Figure 3). *FLC* expression can be suppressed by the autonomous pathway complex, which mainly contains two types of factors, RNA processing mediated by *FPA*, *FY*, *FLK*, and *FCA*, and histone modification mediated by *FLD*, *FVE*, *REF6*, and *LD*. Thus, the up-regulation of *LD*, *REF6*, *FY*, *FPA*, *FCA*, *FLK*, and *FVE* in *mdn1-1* (Figure 2) might partly contribute to *FLC* down-regulation. The expression of *FLC* is indeed increased in *mdn1-1 flk* and *mdn1-1 fve-4* double mutants, which further supports this view (Appendix A). However, knockout of the *FLC* function in *mdn1-1* led to a severely delayed flowering phenotype (see *mdn1-1 flc-3* double mutant in Figure 3), suggesting that FLC is also required for timely flowering in *mdn1-1* (Figure 3A–C). Since FLC is a MADS-box transcription factor, it is likely that there are other factors in *mdn1-1* promoting flowering by forming a complex with FLC, but the *FLC* mutation makes the complex ineffective. This hypothesis needs to be further explored. Additionally, the *mdn1-1 flk* and *mdn1-1 fve-4* double mutants flowered earlier *flk* and *fve-4*, suggesting that in addition to the autonomous-FLC pathway, other flowering pathways might be also involved in *mdn1-1* early flowering (Figure 3D–F and Appendix A). Transcriptomic analysis showed that the expression of several photoperiod pathway genes was changed, implying that photoperiod-pathway genes might play a role in *mdn1-1* early flowering (Figure 2). Furthermore, the flowering time of both *co* and *ft-10* mutants could be accelerated by introducing the *mdn1-1* allele, suggesting that other flowering integrators downstream of CO and FT might play a role in *mdn1-1* early flowering (Figure 4). In addition, the *mdn1-1 clf-28* double mutant bolted earlier than *mdn1-1*, further suggesting that the photoperiod pathway played a significant role in *mdn1-1* early flowering (Figure 5).

The double mutant *mdn1-1 abi5* exhibited a late-flowering phenotype than the *mdn1-1* single mutant, suggesting that the *ABI5* played a positive role in *mdn1-1* early flowering (Figure 6). Nevertheless, the molecular mechanisms by which ABI5 responds to the *MDN1* mutation and regulates flowering in *mdn1-1* remain to be illustrated.

In this study, we found that the mutation of *MDN1* led to early flowering, which is distinctive from previously reported RBF mutants. Transcriptome and genetic analysis suggest that multiple flowering regulation pathways might devote to the early flowering in *mdn1-1*. Intriguingly, we found a positive regulator, ABI5, in response to ribosome biogenesis defect for growth stage switching.

## Figures and Tables

**Figure 1 ijms-23-06509-f001:**
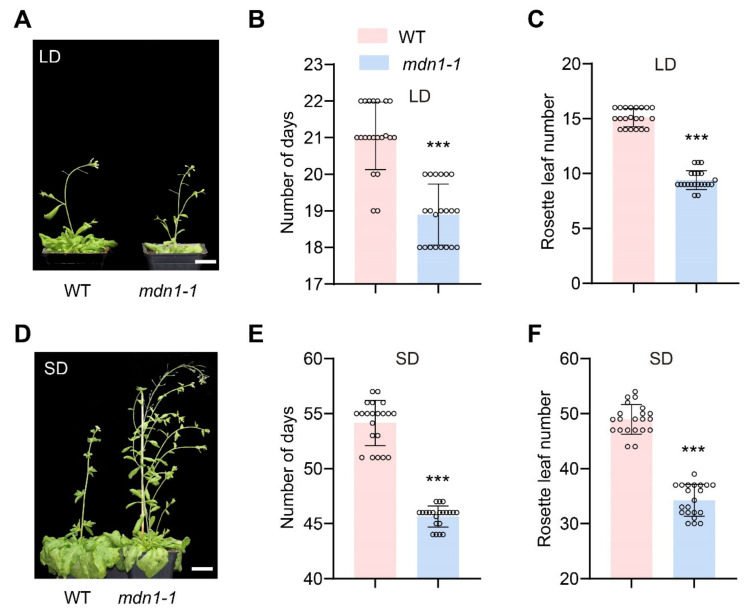
Mutation in *MDN1* leads to early flowering. (**A**)Wild-type (WT) and *mdn1-1* plants grown in long days (LD). (**B**,**C**) The number of days and rosette leaf number at flowering grown in LD. Twenty plants (grown in LD) were scored. (**D**) WT and *mdn1-1* plants grown in short days (SD). (**E**,**F**) The number of days and rosette leaf number at flowering grown in SD. Twenty plants (grown in SD) were scored. (*******
*p* < 0.001, Student’s *t*-test). Scale bars, 2 cm.

**Figure 2 ijms-23-06509-f002:**
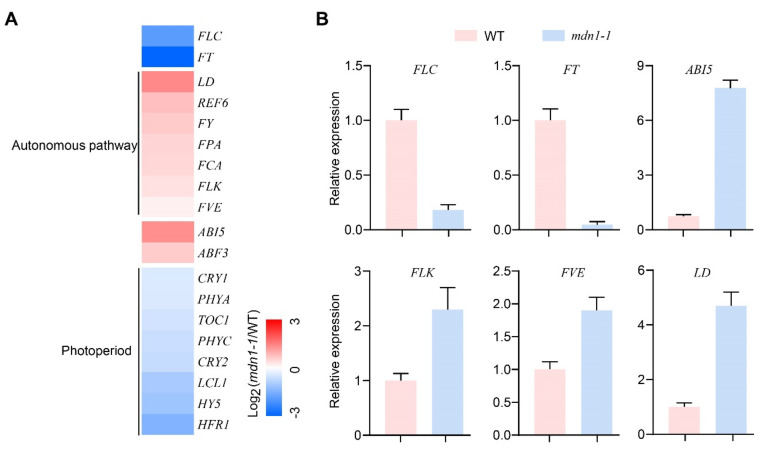
Gene expression profiling of WT and *mdn1-1* seedlings. (**A**) The expression analysis of genes involved in flowering time regulation uses the log_2_-transformed fold-change values. Red, blue, and white indicate an increase, decrease, and no difference in expression levels, respectively. (**B**) Quantitative RT-PCR (qRT-PCR) results showing the expression of the indicated genes in the WT and *mdn1-1* seedlings.

**Figure 3 ijms-23-06509-f003:**
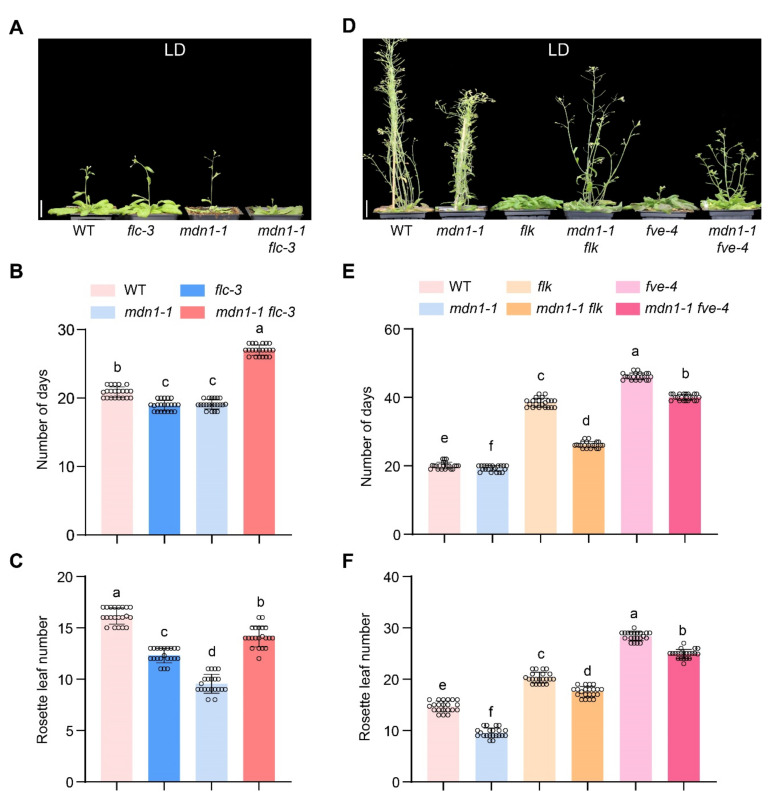
The flowering time of *FLC*-related mutants in *mdn1-1* background. (**A**) *flc-3*, *mdn1-1* and *mdn1-1 flc-3* plants grown in LD. (**B**,**C**) The number of days and rosette leaf number at the flowering in LD. (**D**) *mdn1-1*, *flk*, *mdn1-1 flk*, *fve-4,* and *mdn1-1 fve-4* plants grown in LD. (**E**,**F**) The number of days and rosette leaf number at the flowering in SD. Twenty plants per line were scored. One-way ANOVA was employed, and the lowercase letters denote distinct groups (*p* < 0.01). Scale bars, 2 cm.

**Figure 4 ijms-23-06509-f004:**
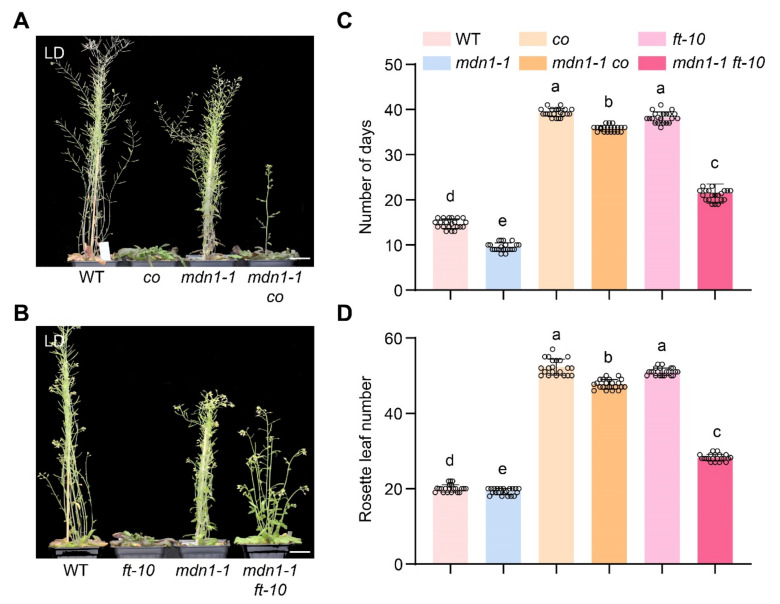
*co* and *ft-10* inhibit the early flowering of *mdn1-1*. (**A**) *co*, *mdn1-1* and *mdn1-1 co* plants grown in LD. (**B**) *ft-10*, *mdn1-1* and *mdn1-1 ft-10* grown in LD. (**C**,**D**) The number of days and rosette leaf number at the flowering in LD. Twenty plants per line were scored. One-way ANOVA was employed, and the lowercase letters denote distinct groups (*p* < 0.01). Scale bars, 2 cm.

**Figure 5 ijms-23-06509-f005:**
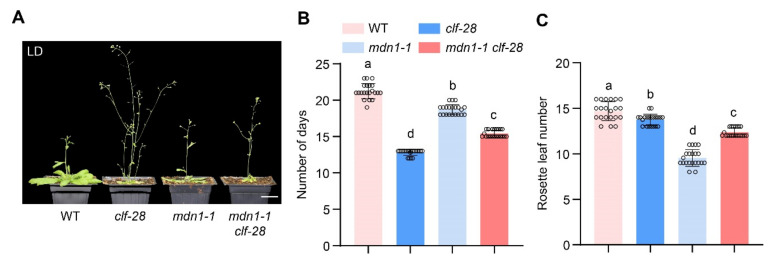
*clf-28* accelerates the early flowering of *mdn1-1*. (**A**) *clf-28*, *mdn1-1* and *mdn1-1 clf-28* grown in LD. (**B**,**C**) The number of days and rosette leaf number at the flowering in LD. Twenty plants per line were scored. One-way ANOVA was employed, and the lowercase letters denote distinct groups (*p* < 0.01). Scale bars, 2 cm.

**Figure 6 ijms-23-06509-f006:**
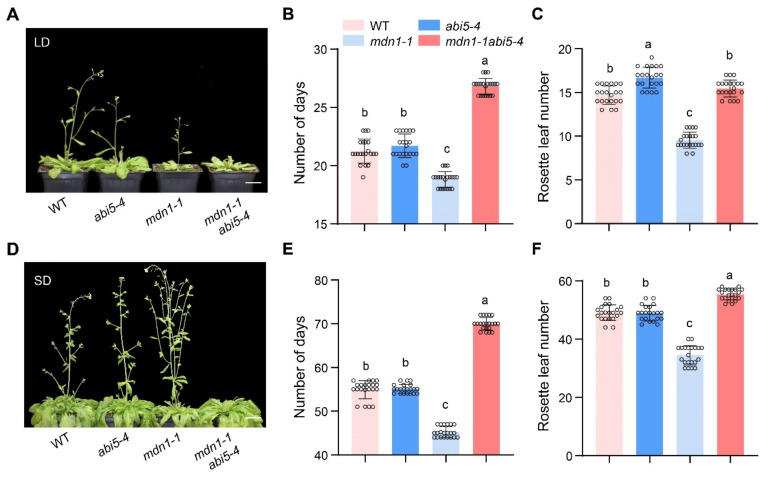
*abi5-4* inhibits the early flowering of *mdn1-1*. (**A**) *abi5-4*, *mdn1-1* and *mdn1-1 abi5-4* grown in LD. (**B**,**C**) The number of days and rosette leaf number at the flowering in LD. (**D**) *abi5-4*, *mdn1-1* and *mdn1-1 abi5-4* grown in SD. (**E**,**F**) The number of days and rosette leaf number at the flowering in SD. Twenty plants per line were scored. One-way ANOVA was employed, and the lowercase letters denote distinct groups (*p* < 0.01). Scale bars, 2 cm.

## Data Availability

Not applicable.

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

*thaliana*. Plant J..

