# Peer review of "Mutation of an Essential 60S Ribosome Assembly Factor MIDASIN 1 Induces Early Flowering in Arabidopsis"

_ijms, 2022, doi:10.3390/ijms23126509_

Round 1
Reviewer 1 Report
In this article, the authors studied the early flowering of mdn1-1 mutant, and the possible relationship of MDN1 and other genes that might together regulating the flowering timing. The finding of this research is very interesting, since most ribosomal protein defects leads to late flowering, while mdn1-1 leads to early flowering. However, there are some aspects making this study not sound for the conclusions. I believe the authors should provide more experimental evidences to support their findings, and also put more thoughts on the discussion.
The main problem I found is the lacking of DNA expression data validated by qRT-PCR experiments. A lot of the experiments designed from the result of RNA-seq data using 7-d seedlings as materials, and I’m not sure if this is a good choice for flowering timing studies. I would like to see more qRT-PCR or at least RT-PCR data of the key genes to support the RNA-seq data, and also for experiments in figure 3-6, to support the authors’ discussion on the possible involvement of genes that were tested.
Other detailed comments:
Line 141, I don't think Table 1 is necessary to be included in the paper, or maybe move to supplemental
Line 149-152, I didn't see data about mdn1-1/mdn1-2, perhaps should be included in supplemental?
173-174, These are not the top 3 enrichement result, why list these not the top 3?
Line 179, where is GA2ox1 in figure 2? I only see GA2ox2 and it seems highly upregulated.
Figure 2, Why GID1A is down-regulated?
How's FVE, Com CLF expression?
Why put FT on the leaves and FLC on top? Any indication on their expression of tissues? And no FT expression mentioned in results.
Line 183-185, the figure didn't show ABF3. I also don’t see PYR1, is it the PYL1 in Fig2?
214-215, are the authors suggesting these autonomous pathway genes are not involved? I feel it’s too early to make that conclusion.
Line 235, where is the bolting time data?
Line 267, and also in the discussion, the conclusion about ABI5 is a bit too early. Please provide more experimental evidences, like the qRT-PCR of ABI5 in mdn1-1, overexpression of ABI5 in WT and mdn1-1, or complementation in abi5-4 and the double mutant.
Line 308-309, I don't see how Fig 4 support this conclusion?
Line 309-311, more discussion on why mdn1-1 flc-3 double mutant shows completely opposite phenotype of the two original single mutant?
Line 319-321, Isn't CLF upstream of FT? I don't understand how is this concluded?
Figure 3B, 6B and 6E, y-axis are not starting from 0!
The authors tested a lot of genes and mutants, and I recognize the hard work behind this. But I still believe the paper needs more experimental evidences to be more profound. Besides, the author might work more on describing the genes and make better links of them in the intro, results, and discussion, making the paper a better story.
Reviewer 2 Report
This manuscript describes the involvement of an 60S ribosome biogenesis factor (MIDASIN 1) to the early flowering of Arabidopsis. A series of SALK mutants were used and the flowering time of the seedlings was estimated. An RNA-seq approach was used to determine the differentially expressed genes between wild type and mdn1-1seedlings.
General comments:
Please specify the purpose of the study. A brief description of the aim of this study is missing.
Some information about the SALK mutants used are needed. Why did you choose these mutants?
Did you confirm the RNA-seq data with any way? A real-time PCR experiment is missing in order to clarify the RNA-seq results.
Minor edits:
Line 9: correct font size
Line 94: erase dot
Line 108 - 128: correct font size
Round 2
Reviewer 1 Report
I appreciate the authors efforts on improving the data and the writing, and explanations on my questions and comments. Most of my questions got explained or revised in the manuscript. I only have 2 minor comments:
1. Line 214-218, I'm glad to see the authors use qRT-PCR data to support the conclusion here, but I still think it's not quite enough. How many replicates has the authors done? I assume less than 3 since I didn't see statistical analysis. The FLC expression seems significantly increased in the flk and fve single and double mutants, but I cannot tell if there's any differences between the flk single vs. double, or fve single vs. double mutants, and therefore it's not enough to explain the early flowering in the double mutants compared with the single flk or fve mutants. If the authors did have 3 replicates and the statistics showing there're no significant differences among them, it might be more like what the authors discussed in the discussion.
2. Line 231-232, I still don't see the bolting time data? Unless the authors mean the flowering time data? I don't think bolting and flowering are exactly the same thing here, so please be more precise in text.
Reviewer 2 Report
Thank you for completing the revisions.
Author Response
We appreciate you very much for the suggestions on our manuscript.